# Factors affecting uptake of the levonorgestrel-releasing intrauterine device: A mixed-method study of social franchise clients in Nigeria

**Aurélie Brunie[1], Anthony Adindu Nwala[2], Kayla Stankevitz[3], Megan Lydon[3], Kendal Danna[4], Kayode Afolabi[5], Kate H. Rademacher[3]***

**1** FHI 360, Washington, DC, United States of America, **2** Society for Family Health, Abuja, Nigeria, **3** FHI 360, Durham, NC, United States of America, **4** Population Services International, Washington, DC, United States of America, **5** Federal Ministry of Health, Abuja, Nigeria

* krademacher@fhi360.org

## Abstract

### Background

Despite the positive characteristics of the levonorgestrel-releasing intrauterine device (IUD)–a long-acting, highly effective contraceptive with important non-contraceptive attributes–the method has not been widely available in low- and middle-income countries. This study of hormonal IUD, copper IUD, implant and injectable users in Nigeria compares their characteristics, reasons for method choice, and experiences obtaining their method.

### Methods

We conducted a phone survey with 888 women who received a hormonal IUD, copper IUD, contraceptive implant or injectable from 40 social franchise clinics across 18 states in Nigeria. We analyzed survey data descriptively by method and assessed factors associated with hormonal IUD use through multivariate logistic regression models. Follow-up in-depth interviews conducted with 32 women were analyzed thematically.

### Results

There were few differences by method used in the socio-demographic profiles and contraceptive history of participants. Among users choosing a long-acting, reversible method, the top reasons for method choice included perceptions that the method was "right for my body," long duration, recommended by provider, recommended by friends/family, few or manageable side effects, and high effectiveness. Among hormonal IUD users, 17% mentioned reduced bleeding (inclusive of lighter, shorter, or no period), and 16% mentioned treatment of heavy or painful periods. Qualitative data supported these findings. Among survey respondents, between 25% and 33% said they would have chosen no method if the method they received had not been available. Both quantitative and qualitative data indicated that partner support can affect contraceptive use, with in-depth interviews revealing

**Data Availability Statement:** The de-identified survey dataset, the survey questionnaire, and the in-depth interview guide are publicly available

online through the Harvard Dataverse (https://dataverse.harvard.edu/dataverse/fhi360_leap). Full qualitative transcripts are not available for ethical reasons because of the risk of deductive disclosure. However, relevant excerpts of transcripts are available from the authors on reasonable request.

**Funding:** KR Investment number: INV-008333 Project title: Learning about Expanded Access and Potential of the LNG-IUS (LEAP) Initiative Funder: Bill & Melinda Gates Foundation https://www.gatesfoundation.org/ The funders had no role in study design, data collection and analysis, decision to publish, or preparation of the manuscript.

**Competing interests:** The authors have declared that no competing interests exist.

that women typically needed partner permission to use contraception, but men were less influential in method choice.

## Conclusions

Expanding access to the hormonal IUD as part of a full method mix provides an opportunity to expand contraceptive choice for women in Nigeria. Findings are timely as the government is poised to introduce the method on a wider scale.

## Introduction

The levonorgestrel intrauterine device (IUD)–also known as the hormonal IUD–is a highly efficacious, long-acting contraceptive. The slow and steady localized release of progestin can minimize systemic side effects, and method use is associated with reduced menstrual blood loss and cramps. The hormonal IUD is a proven treatment for menorrhagia and uterine fibroids and potentially for anemia [1–5]. Yet availability of this method in low- and middle-income countries (LMIC) has previously been limited to small scale distribution in the commercial sector due to the high retail price of existing products. However, the landscape is changing, with more affordable products becoming available [6].

Historically, the prevalence of the copper IUD has been low in many countries in sub-Saharan Africa due to both supply- and demand-side barriers [7], and it is unclear to what degree uptake of the hormonal IUD would face the same challenges given that the two methods share some attributes (e.g., vaginal insertion) but also have key differences (e.g., different bleeding and side effect profiles) [8]. Available research on women's perspectives on the hormonal IUD shows promise for introduction and acceptability of this method in sub-Saharan Africa. A study in a Kenyan clinic where postpartum women were offered the choice between condoms, lactational amenorrhea, pills, injectables, implants, the copper IUD and the hormonal IUD showed robust interest in the hormonal IUD, with 16% of participants opting for this method [9]. Qualitative interviews with hormonal IUD users in Ghana, Kenya and Nigeria all revealed favorable perceptions of the hormonal IUD among women [10–12]. In Ghana, four in five hormonal IUD users recruited from six health facilities expressed satisfaction with the product [10]. Clients of mobile outreach teams, social franchise clinics and public sector facilities supported by Marie Stopes International in Nigeria who were interviewed three months after uptake all intended to continue the method, noting non-contraceptive benefits and fewer side effects compared with other methods [11]. Almost all current or recent hormonal IUD users from two Kenyan clinics reported positive experiences with the method, citing attributes including convenience, few side effects, method duration and reduced periods [12].

Monitoring data from several pilot introduction programs in Kenya, Nigeria, Madagascar, and Zambia indicate that the hormonal IUD is not simply a substitute for other long-acting reversible contraceptives (LARCs), including the copper IUD and implants. While between 25–65% of hormonal IUD acceptors surveyed in these programs told their provider they would have chosen another LARC if the hormonal IUD had not been available, 14–59% would have chosen a short-acting method, and between 3–30% said they would have left with no method. Across most programs, the most common reasons for choosing the hormonal IUD were its effectiveness, long-acting duration, and minimal side effects [13].

To date, little information is available about the user population for the hormonal IUD, what method attributes they find attractive and reasons for selecting the method; existing data

are limited to special populations, qualitative interviews, and service statistics. The research presented in this paper is part of a prospective cohort study with LARC users intended to measure continuation rates. This paper describes the mixed-methods, baseline component of the study to expand the evidence base by providing additional insight into the factors affecting uptake of the method. Objectives specific to the baseline component of the study were to compare women choosing the hormonal IUD to women choosing other methods (copper IUD, implant, and three-month injectable) in terms of their characteristics, reasons for method choice, and experiences obtaining their method.

## Materials and methods

### Study setting

In 2018, modern contraceptive prevalence among all women was 10.5% in Nigeria. IUDs represented 6% of the method mix (7% for married women and 1% for unmarried women). Among married women, the share of IUD is the highest in the South East and South West regions of the country. Almost 80% of IUDs are sourced from the public sector and 20% from the private sector [14]. Recent efforts among both government and non-governmental organizations have been successful in increasing access to family planning including LARCs in Nigeria [15].

The hormonal IUD is not currently available at scale and thus not reflected in national surveys. Since 2007, the International Contraceptive Access (ICA) Foundation, a public-private partnership between Bayer AG and the Population Council, has donated limited quantities of free hormonal IUD devices for distribution by several partners in Nigeria [16]. In 2017, Society for Family Health (SFH) Nigeria began offering donated hormonal IUD units through 40 social franchise clinics in its Healthy Family Network across 18 states spanning all regions of the country. All clinics are located in urban and peri-urban settings and had high pre-existing IUD uptake. At the time of this study, SFH recommended providers charge 3,000 Naira (USD 8.33) for the hormonal IUD, compared to 1,500 Naira (USD 4.17) for implants, 1,000 Naira (USD 2.78) for the copper IUD and 500 Naira (USD 1.39) for injectables.

### Design and data collection

At baseline of this prospective study, we surveyed women opting to receive a hormonal IUD, copper IUD, implant or three-month injectable through one of SFH's 40 social franchise clinics. We conducted a phone survey with a convenience sample of clients within 100 days of method uptake, and follow-up in-depth interviews (IDIs) with a purposive subset of survey participants. Eligible women were clients aged 18–49 who had access to a phone. Family planning providers informed eligible clients and provided the research team with the phone information of those willing to be contacted about the study. We independently confirmed eligibility by extracting the method and date of service from clinic records. Sample size was determined for the prospective study to achieve separate 95% confidence intervals with 5% precision for 12-month continuation rates for each method. We also assumed a 3% intraclass correlation and a 45% loss-to-follow-up rate. Using available information on LARC continuation rates (we used identical assumptions for the copper IUD and the hormonal IUD) [17], we aimed to complete 854 surveys with LARC users at baseline across sites (276 hormonal IUD, 276 copper IUD, 302 implants). We aimed for 50 surveys with injectable users; these were surveyed at baseline only to provide a descriptive comparison of their characteristics to those of LARC users given the popularity of injectables in Nigeria.

We conducted IDIs with a subset of survey participants using a LARC and residing in Oyo (South) or Kaduna (North) state. We purposively selected women to represent a range of

contraceptive history and fertility intentions. Given our focus on the hormonal IUD, we aimed for 12–16 IDIs with hormonal IUD acceptors and 12–16 IDIs with other LARC users based on evidence showing these numbers are sufficient to achieve 80% thematic saturation [18].

The baseline survey covered socio-demographic characteristics, contraceptive history, sources of information about the hormonal IUD, reasons for method choice, potential interest in future hormonal IUD use (for women using other methods), and experiences accessing and receiving services. IDI topics included life goals for the next five years, reproductive and contraceptive history, experiences with previous method, reasons for current method selection, experiences obtaining current method, and perceptions of the hormonal IUD. Questions were informed by the existing literature, and revised through peer review, extensive discussions with local study staff, and field pre-test exercises. Surveys and IDIs were conducted in Pidgin English, Yoruba, or Hausa between June and November 2018. Three research assistants administered the survey questionnaire on the phone by using tablets. Four other research assistants conducted IDIs in-person at a pre-arranged location, including participants' homes or other venues like health facilities. IDIs were audio-recorded and recordings transcribed into English. We obtained verbal consent for the phone survey and written consent for IDIs. Women received 1,000 Naira (USD 2.78) as mobile money for participating in the phone survey and we reimbursed 3,000 Naira (USD 8.33) for transport for IDIs not conducted at home. The National Health Research Ethics Committee in Nigeria and FHI 360's Protection of Human Subjects Committee in the United States approved the study.

## Analysis methods

We compared the characteristics of study participants, reasons for method choice, and experiences obtaining their method descriptively by method received. We then conducted two exploratory multivariable logistic regression analyses to examine the factors associated with uptake of the hormonal IUD relative to the copper IUD and to implants, respectively. We included 12 variables related to socio-demographic characteristics, prior contraceptive use, and partner awareness of baseline method use in all models. We confirmed the absence of multicollinearity using VIF values. Associations were assessed using adjusted odds ratios with their 95% confidence intervals and significance was assessed at the 5% level based on the logistic models. All analyses were done using Stata version 15.

We uploaded IDI transcripts into NVivo 12 for coding and thematic analysis. The codebook combined *a priori* codes driven by the study's objectives and data-driven codes that emerged from the reading of transcripts. Coding was divided between four analysts; intercoder agreement was established at the beginning of the process and verified with approximately 10% of transcripts for ongoing consistency. We developed analytic memos to examine the dimensions of each theme and used Excel matrices to summarize the prevalence of key themes.

## Results

### Survey results

The final sample included 888 women from 39 facilities. From the list of women whom providers referred, 38 women could not be reached and 12 declined participating. We excluded 25 women because they did not meet inclusion criteria upon verification of clinic records.

### Characteristics

The mean age by method ranged from 32 to 34 years old and the mean parity from 3.1 to 3.5 (Table 1). Education level, proportion married, and wealth status of social franchise clients

**Table 1. Participant characteristics.**

| | Hormonal IUD | Copper IUD | Implant | Injectable |
|---|---|---|---|---|
| | n = 266 | n = 274 | n = 295 | n = 53 |
| **Age** | | | | |
| 18–24 | 21 (7.9%) | 12 (4.4%) | 40 (13.6%) | 3 (5.7%) |
| 25–34 | 127 (47.7%) | 130 (47.4%) | 148 (50.2%) | 28 (52.8%) |
| 35–49 | 118 (44.4%) | 132 (48.2%) | 107 (36.3%) | 22 (41.5%) |
| Mean age (SD) | 33.3 (6.1) | 33.7 (5.7) | 31.7 (6.3) | 32.3 (6.1) |
| **Married** | 258 (97.0%) | 262 (96.0%) | 273 (92.5%) | 47 (88.7%) |
| **Highest education completed** | | | | |
| None or some primary | 5 (1.9%) | 10 (3.7%) | 11 (3.7%) | 1 (1.9%) |
| Primary | 31 (11.7%) | 29 (10.5%) | 36 (12.2%) | 10 (18.8%) |
| Secondary | 113 (42.5%) | 127 (46.4%) | 136 (46.1%) | 25 (47.2%) |
| More than secondary | 117 (44.0%) | 108 (39.4%) | 112 (38.0%) | 17 (32.1%) |
| **Parity** | | | | |
| 0 | 5 (1.9%) | 2 (0.7%) | 10 (3.4%) | 3 (5.7%) |
| 1–2 | 74 (27.8%) | 81 (29.6%) | 104 (35.5%) | 15 (28.3%) |
| 3–4 | 137 (51.5%) | 121 (44.2%) | 135 (46.1%) | 27 (50.9%) |
| 5+ | 50 (18.8%) | 70 (25.5%) | 44 (15.0%) | 8 (15.1%) |
| Mean number of children (SD) | 3.3 (1.5) | 3.5 (1.7) | 3.1 (1.7) | 3.1 (1.7) |
| **Fertility intentions** | | | | |
| Child within 2 years/timing undecided | 32 (12.0%) | 51 (18.6%) | 45 (15.3%) | 16 (30.2%) |
| Child in 2+ years | 85 (32.0%) | 64 (23.4%) | 107 (36.3%) | 11 (20.8%) |
| Child, timing undecided | 10 (3.8%) | 8 (2.9%) | 18 (6.1%) | 0 (0.0%) |
| No more children | 93 (35.0%) | 114 (41.6%) | 90 (30.5%) | 17 (32.1%) |
| Undecided about more children | 46 (17.3%) | 37 (13.5%) | 35 (11.9%) | 9 (17.0%) |
| **Urban Wealth quintile[a]** | | | | |
| Lowest | 4 (1.5%) | 5 (1.8%) | 7 (2.4%) | 2 (3.8%) |
| Second | 23 (8.6%) | 16 (5.9%) | 22 (7.5%) | 6 (11.3%) |
| Middle | 29 (10.9%) | 31 (11.4%) | 38 (13.0%) | 8 (15.1%) |
| Fourth | 50 (18.8%) | 54 (19.9%) | 67 (22.9%) | 11 (20.8%) |
| Highest | 160 (60.2%) | 166 (61.0%) | 159 (54.3%) | 26 (49.1%) |
| **Full-time or self-employed** | 197 (74.1%) | 200 (73.0%) | 191 (64.7%) | 37 (69.8%) |

[a] Relative wealth was measured using the equity tool for Nigeria [19]. The urban version of the equity tool compares participants to the urban population in Nigeria.

were high. Compared to women who chose the injectable or the implant, hormonal IUD and copper IUD acceptors were slightly older, more were in the upper urban wealth quintile, and more wanted to limit childbearing.

More injectable users (91%) than LARC users (70–77%) had used a modern method previously (Table 2). Prior use of intrauterine methods was highest among hormonal IUD (20%) and copper IUD (16%) acceptors. Between 55–58% of LARC users and 74% of injectable users reported a short-acting method as their last contraceptive.

## Method choice

Between 25–36% of women who chose other methods had heard about the hormonal IUD prior to the survey. Providers were the most common source of information about the hormonal IUD. Compared to other method users, fewer hormonal IUD acceptors said they already knew they wanted to use the method they received when they went to the clinic (50%

**Table 2. Contraceptive use history and decision making.**

| | Hormonal IUD | Copper IUD | Implant | Injectable |
|---|---|---|---|---|
| | n = 266 | n = 274 | n = 295 | n = 53 |
| **Contraceptive use history** | | | | |
| Ever use of modern contraception[a] | 206 (77.4%) | 208 (75.9%) | 205 (69.5%) | 48 (90.6%) |
| Ever use of intrauterine method | 53 (19.9%) | 44 (16.1%) | 27 (9.2%) | 6 (11.3%) |
| **Last method used** | | | | |
| Hormonal IUD | 5 (1.9%) | 5 (1.8%) | 0 (0.0%) | 0 (0.0%) |
| Copper IUD | 33 (12.4%) | 27 (9.9%) | 12 (4.1%) | 2 (3.8%) |
| IUD- unspecified | 0 (0.0%) | 0 (0.0%) | 3 (1.0%) | 0 (0.0%) |
| Implant | 16 (6.0%) | 16 (5.9%) | 21 (7.1%) | 7 (13.2%) |
| Injectables | 49 (18.4%) | 52 (19.0%) | 54 (18.4%) | 17 (32.1%) |
| Pills | 18 (6.8%) | 29 (10.6%) | 28 (9.5%) | 4 (7.5%) |
| Male condoms | 73 (27.4%) | 62 (22.7%) | 67 (22.8%) | 15 (28.3%) |
| Emergency contraception | 10 (3.8%) | 13 (4.8%) | 16 (5.4%) | 3 (5.7%) |
| Other method[b] | 2 (0.8%) | 3 (1.1%) | 3 (1.0%) | 0 (0.0%) |
| No modern method | 60 (22.6%) | 66 (24.2%) | 90 (30.6%) | 5 (9.4%) |
| **Used modern method in three months prior to receiving method** | 104 (39.1%) | 77 (28.1%) | 88 (29.8%) | 27 (50.9%) |
| **Had heard about hormonal IUD at time of survey** | 266 (100%) | 77 (28.1%) | 75 (25.4%) | 19 (35.8%) |
| **Sources of information about the hormonal IUD[c]** | | | | |
| Provider during visit for method | 207 (77.8%) | 75 (97.4%) | 69 (92.0%) | 17 (89.5%) |
| Provider, other visit or referral | 39 (14.7%) | 3 (1.1%) | 4 (1.4%) | 1 (1.9%) |
| Friends/family | 93 (35.0%) | 3 (3.9%) | 6 (8.0%) | 6 (31.6%) |
| Community volunteer/IPC agent | 44 (16.5%) | 2 (2.6%) | 2 (2.7%) | 2 (10.5%) |
| **Interested in using the hormonal IUD at any time in the future[c]** | N/A | 57 (74.0%) | 50 (66.7%) | 13 (68.4%) |
| **Knew prior to visit that they wanted to use method** | 132 (49.6%) | 163 (59.5%) | 176 (59.7%) | 39 (73.6%) |
| **Made decision without being influenced by others** | 107 (81.1%) | 136 (83.4%) | 146 (83.0%) | 36 (92.3%) |
| **Partner aware of baseline method use** | 236 (89.4%) | 248 (91.2%) | 267 (91.1%) | 43 (81.1%) |

IPC = Interpersonal communication.

[a] For the purposes of this analysis, modern methods include the hormonal IUD, the Copper IUD, implants, injectables, pills, emergency contraception and male and female condoms.

[b] Other methods include female condoms and standard days method.

[c] Among women who had heard about the hormonal IUD. Multiple responses possible.

vs. 60–74%). More injectable users (92%) than LARC users (81–83%) said they chose their method without being influenced by others. Over 89% of LARC users and 81% of injectable users reported that their partner knew they were using their method.

In the multivariable models (Table 3), we did not find any significant association between any of the covariates and hormonal IUD use relative to copper IUD use. Married women and women who had previously used a LARC had higher odds of using the hormonal IUD relative to implants, while women whose partner was aware of method use had higher odds of implant use.

A perception that the method was "right for my body," long duration, recommended by provider, recommended by friends/family, few/manageable side effects, and high effectiveness were cited as reasons for method choice by at least 25% of women across LARCs (Fig 1). Among hormonal IUD users, 17% mentioned reduced bleeding (inclusive of lighter, shorter or no period), and 16% mentioned treatment of heavy or painful period.

**Table 3. Adjusted odds ratio estimates and 95% confidence interval from logistic regression analyses associated with choosing the hormonal IUD over other methods.**

| Characteristic (reference group) | Hormonal IUD vs. copper IUD | Hormonal IUD vs. implant |
|---|---|---|
| | (n = 538) | (n = 557) |
| *Variables* | *OR (95% CI)* | *OR (95% CI)* |
| **Age** | 0.997 (0.961–1.034) | 1.025 (0.988–1.064) |
| **Currently married** | 1.589 (0.574–4.401) | **3.067 (1.097–8.577)** |
| **Completed secondary school** | 1.086 (0.631–1.869) | 1.343 (0.781–2.310) |
| **Wealth categories (1st-3rd)** | | |
| Fourth | 0.803 (0.454–1.422) | 0.751 (0.430–1.310) |
| Highest | 0.835 (0.517–1.348) | 0.937 (0.584–1.505) |
| **Full-time or self-employed** | 1.094 (0.730–1.640) | 1.242 (0.834–1.850) |
| Parity | 0.911 (0.775–1.071) | 0.970 (0.823–1.142) |
| **Fertility intentions (no/no more children)** | | |
| Undecided about having more children | 1.401 (0.808–2.429) | 1.501 (0.853–2.640) |
| More children, in >2 years | 1.449 (0.848–2.476) | 1.065 (0.637–1.778) |
| More children, within 2 years or undecided timing | 0.748 (0.414–1.349) | 0.921 (0.508–1.671) |
| **Prior contraceptive use (never used a modern method)** | | |
| Used any IUD or IUD | 1.479 (0.811–2.697) | **2.927 (1.527–5.612)** |
| Used other modern method | 1.047 (0.668–1.643) | 1.396 (0.907–2.149) |
| **Prior experience of increased bleeding** | 0.834 (0.505–1.377) | 1.331 (0.767–2.310) |
| **Prior experience of reduced bleeding or amenorrhea** | 1.253 (0.697–2.255) | 0.781 (0.451–1.353) |
| **Prior experience of bleeding disturbances** | 1.228 (0.767–1.966) | 0.820 (0.512–1.315) |
| **Partner aware of baseline method use** | 0.707 (0.388–1.289) | **0.512 (0.262–1.001)** |

Age and parity are interval variables. Other variables are yes/no binary variables, with no as the reference level or categorical variables with the reference level included in parentheses. For urban wealth quintiles, the reference level combines the three lowest quintiles. Statistically significant values (p≤ 0.05) are bolded.

When asked what method they would have chosen if the method they received had not been available, between 25% and 33% of women said they would have chosen no method (Fig 2). Among hormonal IUD acceptors, 23% would have chosen implants, 19% a copper IUD, and 14% a short-acting method. Copper IUD users would most commonly have selected an implant (22%), whereas implant users would have primarily selected injectables (16%) or the copper IUD (13%). The most common responses among injectable users were short-acting methods like condoms (13%) or pills (11%).

Among those who were not using a hormonal IUD but had heard about the method (n = 171), 74% of copper IUD users, 67% of implant users and 68% of injectable users said they may be interested in using the hormonal IUD at some point in the future (Table 2). Of the 51 women who said they would not use the hormonal IUD, the main reason, was fear of the insertion procedure.

**Experiences obtaining methods.** Between 85–91% of women recalled being counseled on other methods, and 68% of injectable users and 83–85% of LARC users reported being informed about contraceptive-induced menstrual changes (CIMCs) and/or other non-bleeding side effects (Table 4). Among women who said they were counseled on side effects, the type of CIMCs women most commonly reported being counseled on was bleeding disturbances (defined as changes in frequency, spotting, or irregular periods). When looking at the

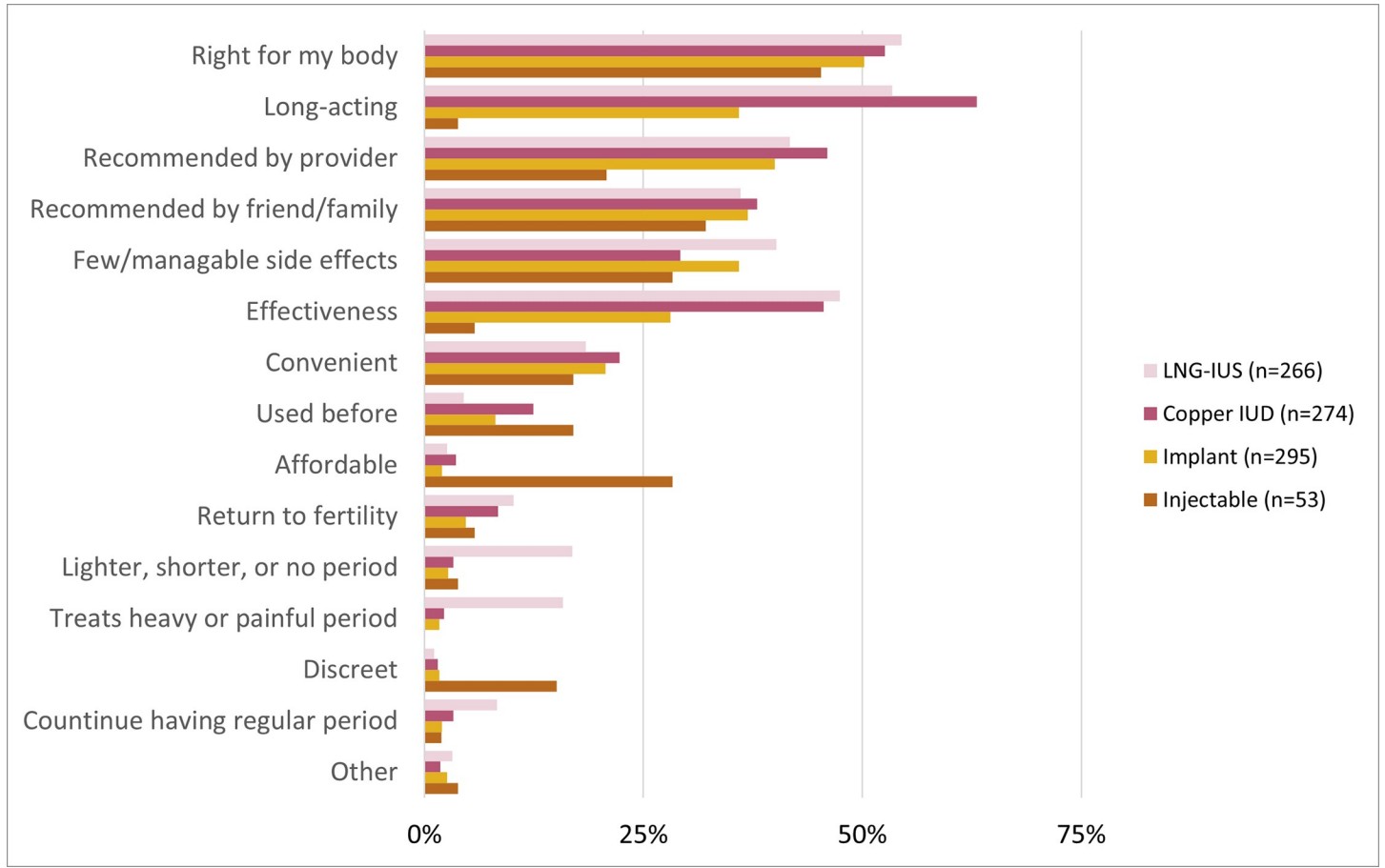

**Fig 1. Reasons for method choice.**

menstrual disruptions that can commonly be expected for each method, among women counseled, 60% of copper IUD users said they were told about heavier or prolonged bleeding, while 48% of hormonal IUD acceptors recalled being told about lighter or shorter bleeding, 29% about amenorrhea, 32% about heavier or prolonged bleeding, and 16% about reduced menstrual pain.

Overall, 90% of hormonal IUD acceptors, and 77–80% of other LARC users correctly reported the duration of their method. Over 95% of LARC users across methods were told by the provider that they could remove their method at any time; however, over half were not told about any other place their method could be removed besides where it was inserted.

Over 92% of women said there was sufficient privacy during the insertion procedure. Just over a quarter (26%) of women who received an implant and 33% of those who received a copper IUD or a hormonal IUD reported experiencing problems when they received the method, compared with 8% of injectable users. Few problems were reported besides temporary pain or discomfort.

Over 78% of women obtained their method during their first visit to the clinic. Reasons for not getting a method the first time included having come to the clinic to get information only, needing partner approval first, or the provider being unavailable.

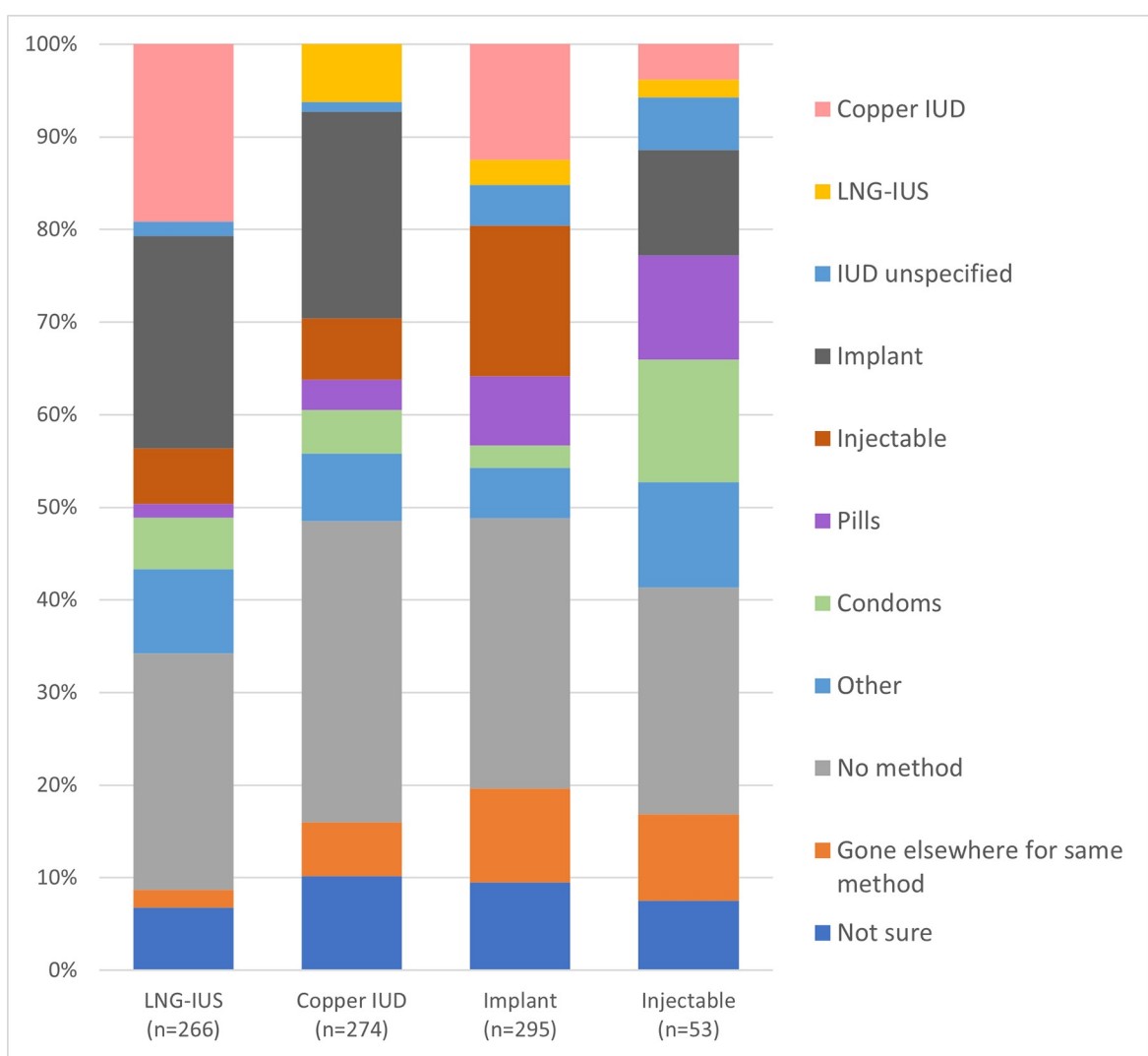

**Fig 2. Method client would have chosen if method received was unavailable.**

## IDI results

There were 32 IDIs, conducted with 17 hormonal IUD, 4 copper IUD and 11 implant acceptors. Most IDIs were conducted within 30 days of the survey, and all within 16 weeks. Here we organized results according to the considerations that led women to choosing a method and to contextual barriers, with particular attention to factors supporting or hindering hormonal IUD uptake. Because reduced menstrual bleeding is a distinctive feature of the hormonal IUD, interviews probed to establish women's perspectives on reduced bleeding and amenorrhea more broadly and findings related to this theme are presented separately.

**Women's considerations related to method choice.** Respondents highlighted similar appealing method characteristics across LARCs. Nearly two-thirds said that the effectiveness and extended duration of their method gave them peace of mind that no pregnancy would occur, allowing them not to worry when having sex, to raise their children properly, and to focus on work. Many women noted the greater reliability and convenience of LARCs over injectables and pills due to reduced user involvement and fewer clinic visits. Several hormonal

**Table 4. Experience with counseling and services.**

| | Hormonal IUD | Copper IUD | Implant | Injectable |
|---|---|---|---|---|
| | n = 266 | n = 274 | n = 295 | n = 53 |
| **Told by provider about other methods** | 239 (89.8%) | 232 (84.7%) | 258 (87.5%) | 48 (90.6%) |
| **Told about CIMCs and/or non-bleeding side effects** | 227 (85.3%) | 230 (83.9%) | 244 (82.7%) | 36 (67.9%) |
| **Non-bleeding side effects mentioned by provider, %** [a] | | | | |
| Weight gain | 87 (38.3%) | 76 (33.0%) | 106 (43.4%) | 13 (36.1%) |
| Abdominal pain | 51 (22.5%) | 43 (18.7%) | 29 (11.9%) | 3 (8.3%) |
| Headaches | 50 (22.0%) | 55 (23.9%) | 87 (35.7%) | 4 (11.1%) |
| Mood changes | 17 (7.5%) | 9 (3.9%) | 21 (8.6%) | 0 (0.0%) |
| Nausea/vomiting | 16 (7.0%) | 23 (10.0%) | 41 (16.8%) | 3 (8.3%) |
| Other | 36 (15.9%) | 54 (23.5%) | 25 (10.2%) | 4 (11.1%) |
| *Any type* | 131 (57.7%) | 138 (60.0%) | 151 (61.9%) | 17 (47.2%) |
| **CIMCs mentioned by provider, %** [a] | | | | |
| Bleeding disturbances | 190 (83.7%) | 180 (78.3%) | 197 (80.7%) | 23 (63.9%) |
| Lighter or shorter bleeding | 108 (47.6%) | 74 (32.2%) | 84 (34.4%) | 5 (13.9%) |
| Heavier or longer bleeding | 72 (31.7%) | 137 (59.6%) | 121 (49.6%) | 13 (36.1%) |
| No bleeding | 66 (29.1%) | 29 (12.6%) | 48 (19.7%) | 10 (27.8%) |
| Less pain during period | 37 (16.3%) | 4 (1.7%) | 0 (0.0%) | 0 (0.0%) |
| *Any type* | 219 (96.5%) | 220 (95.7%) | 227 (93.0%) | 31 (86.1%) |
| **Correctly reported method duration** [b] | 240 (90.2%) | 211 (77.0%) | 210 (79.5%) | 49 (92.5%) |
| **Told by provider at insertion that method can be removed at any time they want** | 260 (97.7%) | 265 (96.7%) | 282 (95.6%) | N/A |
| **Told at insertion where removal can be obtained, %** | | | | |
| Insertion place only | 151 (56.8%) | 150 (54.7%) | 160 (54.2%) | N/A |
| At any clinic | 89 (33.5%) | 103 (37.6%) | 111 (37.6%) | N/A |
| Not told about any place, don't know | 26 (9.8%) | 21 (7.7%) | 24 (8.1%) | N/A |
| **Felt privacy sufficient when received method** | 249 (93.6%) | 258 (94.2%) | 274 (92.9%) | 52 (98.1%) |
| **Experienced problems when received method** | 88 (33.1%) | 91 (33.2%) | 77 (26.1%) | 4 (7.5%) |
| **Median price paid for method (N)** | 3000 | 1500 | 1500 | 500 |
| **Obtained method during first clinic visit** | 210 (78.9%) | 226 (82.5%) | 247 (83.7%) | 46 (86.8%) |

CIMCs = Contraceptive-induced menstrual changes.

[a] Among women counseled on CIMCs and/or non-bleeding side effects. Multiple responses possible.

[b] Based on the duration participants recalled being told by providers when receiving the method. For implants, correct duration was determined based on implant type as informed by participant reports of the number of rods in their implants. Those who did not know their implant type (n = 31) were excluded.

IUD and copper IUD acceptors appreciated that return to fertility would be immediate. A 27-year-old hormonal IUD user said:

> *If I am not carrying a child or pregnant, I have the freedom to work, but if you have a child and you have a baby, you are not so free to work so that is the benefit. . .in five years, I will be able to contribute significantly to my family and even if I get pregnant, I will not be under pressure to buy baby things, and my child will not be looking dirty.*—A223

Finding a method with minimal or tolerable side effects was important in selecting a method for many women, especially hormonal IUD users. Across methods, several women who had a prior negative experience with contraception were motivated to find a different method that would not cause the same problem, while a few others preferred staying on a method that worked well for them. Six of the 17 hormonal IUD acceptors specifically

mentioned reduced bleeding as a motivation for choosing the hormonal IUD. Of these, four had previously experienced heavy bleeding with the copper IUD. Another participant chose the hormonal IUD after experiencing dysmenorrhea with the copper IUD. Several women across methods mentioned that their goal was to minimize effects on menstruation, like this hormonal IUD user who was a first-time contraceptive user: "*I look for options that are as close to the normal thing as possible*" (A111).

Although some participants acknowledged that experiences with a method may vary across women, many indicated that hearing about others' experience–typically close friends or relatives—was a deciding factor, as explained by this 30-year-old hormonal IUD user:

*I saw others doing it and none of them had any complaint of headache or anything. So, I said, okay, let me do something that will give me peace.*–A133

Among hormonal IUD users specifically, several women reporting earlier challenges with side effects or method effectiveness explained they selected the hormonal IUD after the provider recommended it as a better option. A few other hormonal IUD users explained they were convinced to take up the method based on the information received from the provider. Several hormonal IUD users explained they decided to give the method "a try," as explained by this 33-year-old woman who started the method a few months after giving birth to her third child:

*It was the nurse that told me about it. She told me that they have brought a new [method] and that it may be better than the other ones. Then, I said, okay, let me just use it and see.*– A122

Several participants, including some implant users but also a few hormonal IUD acceptors, expressed concerns around intrauterine placement, including dislike of exposing themselves and not wanting their sexual partner to feel strings during sex, as well as fear of damage inside the womb, of experiencing pain during sex, or of migration to the stomach. At the same time, some women reported they did not like injections. A few women who chose intrauterine methods said that this mode of delivery was more discreet, caused less pain, or felt it was less invasive than inserting an implant.

**Contextual barriers.** Most women, especially hormonal IUD acceptors, acknowledged needing partner permission to use family planning. Many women indicated that their partner agreed to a specific method, while some women said their partners had no opinion and a few women's partners recommended the method themselves. Several participants reported discussing side effects with their partner when seeking permission or evaluating methods together, including this 42-year-old hormonal IUD user who was previously experiencing heavy bleeding with a copper IUD:

*[I talked with] my husband. . .because I have to tell him anything I want to do before I do it. . .. He said I can change [my copper IUD] because he knows I was having challenges, so immediately I said I want to change it. He said, okay, I should go and change it.*–A132

More hormonal IUD users than copper IUD and implant users in the sample did not obtain their method the first time they went to the clinic. Among hormonal IUD users who returned a second time, several indicated they had not brought enough money to the initial visit, and a few explained that they needed to discuss with their partner. Two hormonal IUD users and one implant acceptor reported they had to return due to lack of equipment or staff at the

facility. Many hormonal IUD users and one copper IUD acceptor indicated receiving financial support from their partner to pay for their method, including two whose husbands accompanied them to the clinic.

**Perspectives on reduced bleeding and amenorrhea.** IDIs discussed the acceptability of reduced bleeding and amenorrhea through women's reports of their experiences and direct probing around perceptions of lighter bleeding and amenorrhea. Twenty-six of the 32 participants expressed concerns related to amenorrhea. While several women said they were not worried about or saw advantages to not menstruating, some of them expressed concerns about amenorrhea in the same interview. Similarly, when asked specifically if they would prefer to experience reduced bleeding or no bleeding, twice as many women favored reduced bleeding while others contradicted themselves. Underlying perceptions of amenorrhea were considerations related to general health and pregnancy. The most common concern was related to "dirt" accumulating in the body in the absence of a period to flush it out. Several women perceived that not having a period was unnatural, contrary to the will of God, or would interfere with "feeling like a woman." Many said that being amenorrhoeic would cause them to be worried about pregnancy status. Additionally, a few women were concerned that amenorrhea could affect their fertility, but a similar number said they knew fertility would return after discontinuing method use. Illustrating the contradictory nature of statements made by some women, one implant user said:

> *Some women did the family planning and did not see their period and they are living okay. So, I will be okay with it if I don't see my period. . .to me, it's not okay for [a] woman not to see her period. A woman is supposed to be seeing her period every month. Because I might feel the thing is affecting your stomach somewhere.–R113*

The perceived or experienced benefits that participants mentioned in relation to amenorrhea were similar to those they associated with reduced bleeding. These included lifestyle improvements ranging from better menstrual hygiene management (e.g., using fewer or no pads and not having to wash blood) to greater freedom. One hormonal IUD user who previously used baby diapers to manage heavy bleeding with the copper IUD said:

> *The money for that pampers now, I don't buy pampers again because I will not use it. . .I can save the money, so it's important because I can save something for my business [frying food but saving to open a shop]. . . that time that I was using [pampers] during my heavy flow, I will not go out for my business but now I am, I can go at any time any day, every day I used to go and I save some money.–A131*

Notably, some factors emerged from the interviews that appeared to influence perceptions of amenorrhea. A few hormonal IUD acceptors who did not want any more children explained that their concerns about amenorrhea were reduced now that they were done with childbearing. A few women also explained that they were not bothered by amenorrhea experienced while using contraception postpartum because they were still breastfeeding or had given birth less than six months earlier and felt it was natural.

## Discussion

This study among social franchise clients across 18 states in Nigeria contributes information on the user population for the hormonal IUD relative to other LARCs and injectables, and on the factors affecting uptake of this method. Together with condoms, implants and injectables lead the method mix in Nigeria, especially among married women, while copper IUD use is

more limited and population surveys do not report on the hormonal IUD [14]. Altogether, characteristics of women in our sample were generally consistent with those of contraceptive users in Nigeria, where contraceptive use is more common among women who are married, have 3–4 children, are more educated, and have greater household wealth [14]. Among adopters of the four contraceptive methods, we generally found few differences in the socio-demographic profile and contraceptive history of participants. Multivariable results did not reveal significant differences between copper IUD and hormonal IUD users. Hormonal IUD users were more likely to be married, to have prior experience with LARCs and to use their method unbeknownst to their partner compared to implant users. About a quarter of clients who chose each LARC were first-time users of modern contraception indicating the strong potential of all LARCs, including the hormonal IUD, to reach new users in the study setting.

Findings reaffirm the importance of offering a robust contraceptive mix that allows women to choose a method based on their preferences. Between a quarter and a third of women would have walked away without a method if their preferred option had not been available. Regarding the hormonal IUD specifically, our findings are similar to other results showing that the hormonal IUD does not just substitute for other LARCs [9]. Rather, many women would have chosen a short-acting method or not taken any method had the hormonal IUD not been available.

Quantitative and qualitative results indicate that the features driving women to choose the hormonal IUD reside in a combination of characteristics shared with other LARCs as well as distinct elements. Appealing characteristics that were shared with other LARCs include duration of protection against pregnancy and very high effectiveness. LARCs also afforded additional convenience linked to reduced clinic visits and user involvement. Although injectables are generally very effective, that was not a common motivator for their use in this study. Conversely, discreetness and affordability were disproportionately given as reasons for choosing injectables compared to LARCs. Notably, between 10 and 20% of LARC users reported wanting another child within the next two years, signaling that the benefits of LARCs attract women who may not intend to use them until their labeled duration of use. Supporting this notion are qualitative findings describing women's desire to delay pregnancy in order to have more freedom to focus on the children they have and/or work.

Unique features of the hormonal IUD that contributed to uptake among users include potential for reduced side effects, reduced or paused bleeding, and treatment for menorrhagia. While both quantitative and qualitative findings showed that these elements influenced method choice, they generally did not figure as prominently in the results as the characteristics shared with other LARCs. Several factors may explain this. First, method selection is heavily informed by a combination of information received from other women and personal experience. To date, however, the number of users and, consequently, the availability of experiential input on the hormonal IUD are limited. In the future, enabling women to speak with champion users could prove a valuable strategy. Second, while women's perceptions of reduced bleeding are favorable, concerns over amenorrhea persist, including beliefs around blood accumulation in the womb and fear that absence of bleeding is a sign of pregnancy. This is consistent with what has been reported elsewhere [20]. Third, our findings uncovered possible gaps and inconsistencies in counseling on CIMCs and non-bleeding side effects. A sizable minority of women did not recall being counseled on side effects, especially among injectable users, or about common side effects of the method they received. Examples include the possibility of lighter or shorter bleeding with the hormonal IUD or of heavier or longer bleeding with the copper IUD. Additionally, some of the side effects clients reported being counseled on are not typical of their method, such as weight gain for the copper IUD. Counseling tools, including client-facing adaptations, could be valuable to increase knowledge of potential bleeding

disruptions among women. One example is the NORMAL job aid, which includes a description of the types of CIMCs that may be expected over time with each method [21]. Importantly, our results also point to the need to address both health and lifestyle implications in messaging. One encouraging finding is the evidence of communication on when and where removal can be procured for LARCs, including the possibility of removing a method prior to its expiration date. Access to removal services is an essential component of rights-based family planning that is receiving increasing attention, especially given unprecedented growth in implant use in sub-Saharan Africa [22–24].

Previous evidence suggests that low uptake of the copper IUD is often due to both provider-side barriers (e.g., lack of comfort or willingness to insert the method) and client-side concerns (e.g., persistent myths and misperceptions) [7]. Fears among potential users related to uterine placement have been shown to constrain copper IUD uptake, which may be a barrier to hormonal IUD uptake as well [25]. While fear of insertion was the main deterrent for women who said they would not use the hormonal IUD at any point in the future, over two thirds of the women who received other methods expressed interest in using the hormonal IUD in the future. Moreover, some women in our qualitative sample were hormonal IUD users who had overcome initial fears surrounding insertion and other aspects of intrauterine placement. Although possible courtesy bias should be kept in mind, taken together, these results indicate that intrauterine placement is not an insurmountable barrier to uptake of the hormonal IUD.

Partner support can affect contraceptive use [26, 27]. Although the methods in this study have potential for discreet use, partner awareness of use was high across the four methods, especially for injectables. In the multivariable analyses, hormonal IUD users were more likely than implant users to use their method discreetly. Qualitative results indicate that while women typically needed partner permission to use contraception, men were less influential in method choice. Financial support by partners was also important, as has been found in other research [28].

Findings highlight lack of awareness of the method as a potential barrier to hormonal IUD uptake. This was similarly identified as a constraint by providers in a recent assessment across five service delivery settings in Nigeria [29]. As demand generation efforts have remained limited to date, knowledge of and demand for the hormonal IUD are largely provider-driven. In this study, the majority of women who chose other methods had not heard about the hormonal IUD. While recall issues have to be considered, given that the method was available at all service delivery points from which women were recruited, this may suggest either gaps in counseling on the full range of available methods or conflation between the copper IUD and the hormonal IUD.

## Limitations

The study population is limited to social franchise clients with phones; education and wealth may be higher than in other settings. Most women in this study were over the age of 25. Although some hormonal IUD introduction efforts have been successful in reaching younger women, more research is needed with younger populations. Social franchise clinics prioritized for hormonal IUD introduction by SFH had high copper IUD uptake and were typically located in urban and peri-urban settings [13]. While this survey of clients across 18 states provides broad geographic and cultural representation, the setting may be particularly conducive to introduction of the hormonal IUD and findings may not easily transfer to other contexts. Recruitment through providers and the requirement that women have and share phone contact details carry a risk of selection bias. The analysis does not differentiate between types of

implants. Survey questions do not allow reporting on the method information index as a measure of quality. Our study design does not enable us to determine if some women failed to get a method due to inability to pay for the hormonal IUD.

## Conclusions

Broader introduction of the hormonal IUD provides an opportunity to expand LARC choices for women in Nigeria, and the government in Nigeria currently plans to scale up the method including in the public sector. Characteristics shared with other LARCs were attractive; unique distinctive features of the hormonal IUD are appealing but potentially less well-known. Limited awareness of the method and gaps in counseling are barriers that need to be addressed to realize the full potential of the method.

## Acknowledgments

The study team thanks Ms. Aderonke Popoola and Mr. Ekerette Emmanuel Udoh for support to study implementation. We acknowledge the contributions of Ms. Claire Brennan to the analysis of qualitative data. We are also thankful to Ms. Samantha Archie for her help in verifying study results. Special acknowledgment goes to Dr. Jennifer Anyanti, Dr. Oluwole Fajemisin, and Dr. Hadiza Khamofu for supporting this study. We thank Dr. Laneta Dorflinger, Dr. Theresa Hoke and Dr. Mario Chen for their careful review of this manuscript.

## Author Contributions

**Conceptualization:** Aurélie Brunie, Kate H. Rademacher.

**Formal analysis:** Aurélie Brunie, Kayla Stankevitz, Megan Lydon.

**Funding acquisition:** Aurélie Brunie, Kate H. Rademacher.

**Investigation:** Aurélie Brunie, Anthony Adindu Nwala, Kayla Stankevitz, Megan Lydon, Kendal Danna, Kayode Afolabi.

**Methodology:** Aurélie Brunie, Anthony Adindu Nwala.

**Project administration:** Aurélie Brunie, Anthony Adindu Nwala, Kendal Danna, Kate H. Rademacher.

**Writing – original draft:** Aurélie Brunie, Kate H. Rademacher.

**Writing – review & editing:** Anthony Adindu Nwala, Kayla Stankevitz, Megan Lydon, Kendal Danna, Kayode Afolabi.

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
