## [Decision Letter · Decision Letter 0]

22 Jul 2021

PONE-D-21-10209

Factors affecting uptake of the levonorgestrel intrauterine system: A mixed-method study of social franchise clients in Nigeria

PLOS ONE

Dear Dr. Rademacher,

Thank you for submitting your manuscript to PLOS ONE. After careful consideration, we feel that it has merit but does not fully meet PLOS ONE’s publication criteria as it currently stands. Therefore, we invite you to submit a revised version of the manuscript that addresses the points raised during the review process.

We look forward to receiving your revised manuscript.

Kind regards,

Federico Ferrari

Academic Editor

PLOS ONE

Journal Requirements:

3. Please include additional information regarding the survey or questionnaire used in the study and ensure that you have provided sufficient details that others could replicate the analyses. For instance, if you developed a questionnaire as part of this study and it is not under a copyright more restrictive than CC-BY, please include a copy, in both the original language and English, as Supporting Information.  If the original language is written in non-Latin characters, for example Amharic, Chinese, or Korean, please use a file format that ensures these characters are visible.

4. Please state whether you validated the questionnaire prior to testing on study participants. Please provide details regarding the validation group within the methods section.

5. Please include a copy of the interview guide used in the study, in both the original language and English, as Supporting Information, or include a citation if it has been published previously.

Reviewers' comments:

Reviewer's Responses to Questions

**Comments to the Author**

1. Is the manuscript technically sound, and do the data support the conclusions?

Reviewer #1: Yes

Reviewer #2: Yes

2. Has the statistical analysis been performed appropriately and rigorously? 

Reviewer #1: Yes

Reviewer #2: Yes

3. Have the authors made all data underlying the findings in their manuscript fully available?

Reviewer #1: No

Reviewer #2: Yes

4. Is the manuscript presented in an intelligible fashion and written in standard English?

Reviewer #1: Yes

Reviewer #2: Yes

5. Review Comments to the Author

Reviewer #1: Excellent article and important contribution to the literature. Recommend acceptance with just a few comments.

1. Study setting: It would be helpful to provide more information about the location of the 18 states where SFH has been distributing hormonal IUS - which regions of the country?

2. Were injectable users all using DMPA-IM? Probably worth noting availability of DMPA-SC/SI at the time of the study as a background contextual factor.

3. I'm surprised that assumed continuation rates for the copper IUS and the LNG IUS are the same, is that really widely borne out in research?

4. The fact that a lower proportion of injectable users were told about bleeding changes (and really, side effects generally) seems worth mentioning somewhere in terms of programmatic implications, especially given all the literature on the importance of counseling on this issue. Or do you think it's because many were continuing/previous injectable users who were already familiar?

5. Discussion: "Appealing characteristics that were shared with other LARCs but not injectables include duration of protection against pregnancy and very high effectiveness." In order to avoid misunderstandings/misperceptions among readers, I wonder if it's worth adding something like "(although injectables are generally very effective, that did not seem to be a motivator for their use in this study") or something along those lines. I also wonder if it's worth noting that affordability and discretion were downsides for LARCs in the study in terms of the importance of reinforcing the importance of method mix.

6. Curious why demand generation efforts have been so limited, can you briefly explain?

7. Probably worth mentioning the results on removal in the discussion and what these results add to that important area of work on LARCs.

8. The age profile of the population in this study is an important limitation, and could reinforce misperceptions that LARCs are not good choices for AGYW. Please add to limitations and flag as an important area for research and evidence generation in settings like Nigeria (or link to research/evidence that exists).

Reviewer #2: Review: Factors affecting update of hormonal IUS in Nigeria PLOS ONE

This study captures information on the socio-demographic profile and experiences of women receiving LARCs or Injectables at private facilities in Nigeria, combining quantitative and qualitative methods. The study aims to provide a quantitative comparison of Hormonal IUS users as compared to other LARC users, and then probes for particular details on experiences before and after the method was accepted.

The study is sound, if limited in its generalizability. Some requests for revision are provided.

1) Line 61-64: The summary of findings form 4 other studies, grouped together with wide ranges in the values, is confusing. Present it in more detail.

2) Line 104-106: Upon reading the results, I expected to see statistical comparisons between LARC and injectable users. Going back to methods, I see why this may not be possible. But in this case, it seems that there is not enough emphasis on how or if LARC users in this study are similar to, or different from, other contraceptive users in Nigeria. How do the injectable users at the franchised facilities compare to other urban contraceptive users, as per DHS data? How are the clients of these facilities similar to or different from other contraceptive users in Nigeria? This context is important – can be provided in intro, or discussion.

3) Line 148: Specify that means are presented by method, else we would expect a single mean.

4) Table 1: Mean(SD) for age is not a percentage. The EquityTool is written thus.

5) Table 3: Consider using age categories. Wealth category descriptions are misleading – rename ‘middle’ to 4th, lowest to (1st-3rd).

6) Table 4: Is it possible from the data to calculate the Method Information Index? If so, this should be reported. If not, add as limitation. Further, please comment further in discussion on how some of the side effects and bleeding changes reported by clients in relation to the method they received are not considered as known side effects of that method (ie – weight gain w copper IUD).

7) Line 279: Do you have any information in the IDIs regarding cost of the method as a criteria for selection? Given that the hormonal IUS used by SFH were donated commodities, can you further describe or reflect on the implications of their pricing strategy? Why would they be priced at 3x that of the copper IUD, when, presumably, the profit for the donated commodity would be greater even if priced equally? How does price influence use? What does that mean for Nigeria’s public sector strategy?

8) Line 334-335: This seems to be a good place to provide context, comparing study participant profile with that of users from DHS data.

9) Line 345-346: Check grammar.

10) Line 366-369: Consider explicitly commenting on provider knowledge related to LARCs, and reference other literature in this area.

11) Line 382-386: Greater discreet use for the more expensive method, and financial support from partners, seem in contradiction. Elaborate.

6. PLOS authors have the option to publish the peer review history of their article (what does this mean?). If published, this will include your full peer review and any attached files.

Reviewer #1: No

Reviewer #2: **Yes: **Nirali M Chakraborty

---

## [Author Response · Author response to Decision Letter 0]

3 Sep 2021

Responses to reviewer comments

Submission: “Factors affecting uptake of the levonorgestrel intrauterine system: A mixed-method study of social franchise clients in Nigeria” - PONE-D-21-10209

Other

• Editor - Data availability statement: The data should be provided as part of the manuscript or its supporting information, or deposited to a public repository.

Thank you. We have added a data availability statement.

• Editor - Please ensure that you include a title page within your main document. We do appreciate that you have a title page document uploaded as a separate file, however, as per our author guidelines (http://journals.plos.org/plosone/s/submission-guidelines#loc-title-page) we do require this to be part of the manuscript file itself and not uploaded separately. Could you therefore please include the title page into the beginning of your manuscript file itself, listing all authors and affiliations.

The title page was included in the manuscript file.

• Please note: The World Health Organization released an announcement about the nomenclature for the levonorgestrel-releasing intrauterine device after the paper was first submitted. We changed the name to hormonal IUD throughout the paper to align with this guidance. 

Introduction

• R2 - Line 61-64: The summary of findings form 4 other studies, grouped together with wide ranges in the values, is confusing. Present it in more detail.

We added more details about the findings from each study.

Methods

• R1 - Study setting: It would be helpful to provide more information about the location of the 18 states where SFH has been distributing hormonal IUS - which regions of the country? 

The 18 states in which the SFH facilities span all regions in the country. They include Kaduna, Kano, Katsina (North West), Gombe, Taraba (North East), Benue, Niger, FCT (North Central), Lagos, Ogun, Oyo, (South West), Abia, Enugu (South East) and Cross River, Akwa Ibom, Edo, Delta and Rivers (South South). We added that the states span all regions of the country.

• R1 - Were injectable users all using DMPA-IM? Probably worth noting availability of DMPA-SC/SI at the time of the study as a background contextual factor.

Thank you for this question. Only DMPA-IM was available from the study sites when the study was conducted. 

• R1 - I'm surprised that assumed continuation rates for the copper IUS and the LNG IUS are the same, is that really widely borne out in research?

There was very little information on continuation rates for the hormonal IUD compared to other methods in sub-Saharan Africa when we designed the study. A cohort study of postpartum women in Kenya found no statistically significant differences between 12-month continuation rates among hormonal IUD and implant users (89% and 92%, respectively); results for the Cu-IUD were not reported due to low uptake. In a prospective cohort study in the United States, the hormonal IUD had a one-year continuation rate of 87%, compared with 84% for the Cu-IUD and 82% for the three-year subdermal implant. The longitudinal results from the present study and a companion study conducted in Zambia were just published in The Lancet Global Health. Twelve-month continuation rates were 87% for the hormonal IUS and copper IUD, and 85% for implants in Nigeria. In Zambia, they were 95% for the hormonal IUS, 89% for the copper IUD, and 83% for implants. In Zambia, time-to-event curves for the implant and the hormonal IUS were statistically different. We did not find any statistically significant differences between the hormonal IUD and the Cu-IUD. 

• Editor - Please state whether you validated the questionnaire prior to testing on study participants. Please provide details regarding the validation group within the methods section.

Survey questionnaires were informed by a review of the existing literature and jointly developed by study staff in the US and Nigeria, after which they underwent independent peer review by FHI 360 staff. Prior to each data collection round, the questionnaires were discussed extensively and tested through role play during data collector training. We made adjustments to the wording of the questions and the pre-coded responses in all local languages based on the feedback of study supervisors and data collectors working directly with similar populations. We then conducted a field pre-test exercise with a population exhibiting similar characteristics to those of study participant groups to inform final adjustments. We similarly pre-tested qualitative topic guides and used the information from pre-test qualitative interviews to ensure that all important themes coming in the responses were captured in the survey pre-coded responses. We added a sentence describing this process to the methods section.

• Editor - Please include additional information regarding the survey or questionnaire used in the study and ensure that you have provided sufficient details that others could replicate the analyses. For instance, if you developed a questionnaire as part of this study and it is not under a copyright more restrictive than CC-BY, please include a copy, in both the original language and English, as Supporting Information. 

• Editor - Please include a copy of the interview guide used in the study, in both the original language and English, as Supporting Information, or include a citation if it has been published previously.

Thank you. The survey and interview guide are publicly available online through the Harvard Dataverse. We indicated this as part of the data availability statement. The questionnaires were developed in English; translated versions are not available as Word document as they were entered directly in the ODK programs.

Results

• R2 - Line 148: Specify that means are presented by method, else we would expect a single mean.

Thank you. We made this edit.

• R2 - Table 1: Mean(SD) for age is not a percentage. The EquityTool is written thus.

Thank you. We edited Table 1.

• R2 - Table 3: Consider using age categories. Wealth category descriptions are misleading – rename ‘middle’ to 4th, lowest to (1st-3rd).

Thank you. We considered using age categories, but ultimately decided against it because we had few people in the lowest and smallest age groups. We have renamed the Wealth categories as suggested.

• R2 - Table 4: Is it possible from the data to calculate the Method Information Index? If so, this should be reported. If not, add as limitation. 

The survey collected information on whether clients were informed about other methods and whether they were told about side effects. However, we did not collect information on whether clients were told what do if they experienced side effects. Therefore, we are not able to report on the MII. This was added to limitations.

• R2 - Line 279: Do you have any information in the IDIs regarding cost of the method as a criteria for selection? Given that the hormonal IUS used by SFH were donated commodities, can you further describe or reflect on the implications of their pricing strategy? Why would they be priced at 3x that of the copper IUD, when, presumably, the profit for the donated commodity would be greater even if priced equally? How does price influence use? What does that mean for Nigeria’s public sector strategy?

Thank you for this insightful question. As indicated in the manuscript, several participants described the need to go collect the necessary funds with which they would return to the health facility another day, but cost was not a major theme described by IDI participants relating to whether or not to select the method. More broadly, the reason for the higher price for the hormonal IUD in the pilot setting was that SFH anticipated that the method would ultimately be scaled up as a “premium product” given commodity costs for commercial products at the time. Therefore, they wanted to start building the market with this in mind. However, since that time, the hormonal IUD has been added to the UNFPA and USAID catalogs at a more affordable public sector price; as such, the ultimate pricing structure in social franchise settings is still being determined in consultation with suppliers and donors. We feel that this level of detail is not appropriate for this manuscript but we could add a footnote with this background information if the editors request it. 

Discussion

• R1 - The fact that a lower proportion of injectable users were told about bleeding changes (and really, side effects generally) seems worth mentioning somewhere in terms of programmatic implications, especially given all the literature on the importance of counseling on this issue. Or do you think it's because many were continuing/previous injectable users who were already familiar?

• R2 - Further, please comment further in discussion on how some of the side effects and bleeding changes reported by clients in relation to the method they received are not considered as known side effects of that method (ie – weight gain w copper IUD).

We expanded this section of the discussion.

• R2 - Line 366-369: Consider explicitly commenting on provider knowledge related to LARCs, and reference other literature in this area. 

Thank you. We added a comment about provider-side barriers to copper IUD provision which may be applicable to hormonal IUD provision as well. 

• R1 - Discussion: "Appealing characteristics that were shared with other LARCs but not injectables include duration of protection against pregnancy and very high effectiveness." In order to avoid misunderstandings/misperceptions among readers, I wonder if it's worth adding something like "(although injectables are generally very effective, that did not seem to be a motivator for their use in this study") or something along those lines. I also wonder if it's worth noting that affordability and discretion were downsides for LARCs in the study in terms of the importance of reinforcing the importance of method mix.

We added these points.

• R1 - Curious why demand generation efforts have been so limited, can you briefly explain?

For demand creation, SFH used a provider-initiated awareness generation model. This involved the provider carrying out talks about contraception with women in the facilities who came for postnatal or child wellness visits. In these contexts, the provider would include the hormonal IUD in the context of a full method mix; there were not dedicated demand creation activities focused solely on this method. The approach is described further in a 2020 publication that is cited in this paper; see Brunie A, Rademacher KH, Nwala AA, Danna K, Saleh M, Afolabi K. Provision of the levonorgestrel intrauterine system in Nigeria: Provider perspectives and service delivery costs. Gates Open Res. 2020;4.

• R1 - Probably worth mentioning the results on removal in the discussion and what these results add to that important area of work on LARCs.

We added this to the discussion.

• R1 - The age profile of the population in this study is an important limitation, and could reinforce misperceptions that LARCs are not good choices for AGYW. Please add to limitations and flag as an important area for research and evidence generation in settings like Nigeria (or link to research/evidence that exists).

• R2 - Line 104-106: Upon reading the results, I expected to see statistical comparisons between LARC and injectable users. Going back to methods, I see why this may not be possible. But in this case, it seems that there is not enough emphasis on how or if LARC users in this study are similar to, or different from, other contraceptive users in Nigeria. How do the injectable users at the franchised facilities compare to other urban contraceptive users, as per DHS data? How are the clients of these facilities similar to or different from other contraceptive users in Nigeria? This context is important – can be provided in intro, or discussion.

• R2 - Line 334-335: This seems to be a good place to provide context, comparing study participant profile with that of users from DHS data.

We added some information on the profile of contraceptive users in Nigeria in the discussion section. As noted under limitations, we also agree that the generalizability of our findings is likely to be limited because the study is focused on social franchise clients with phones. We expect that education and wealth among study participants may be higher than in other settings. We added a brief discussion of the age profile to the limitations.

• R2 - Line 345-346: Check grammar.

Thank you; this has been addressed.

• R2 - Line 382-386: Greater discreet use for the more expensive method, and financial support from partners, seem in contradiction. Elaborate.

While discreet use was significant in the model comparing the hormonal IUD to implants, as shown in Table 2, male partners were by and large aware of method use. Thus we do not believe that this finding contradicts the one on financial support from partners.

---

## [Editor Report · Decision Letter 1]

10 Sep 2021

Factors affecting uptake of the levonorgestrel-releasing intrauterine device: A mixed-method study of social franchise clients in Nigeria

PONE-D-21-10209R1

Dear Dr. Rademacher,

We’re pleased to inform you that your manuscript has been judged scientifically suitable for publication and will be formally accepted for publication once it meets all outstanding technical requirements.

Kind regards,

Federico Ferrari

Academic Editor

PLOS ONE
---

## [Editor Report · Acceptance letter]

20 Sep 2021

PONE-D-21-10209R1 

Factors affecting uptake of the levonorgestrel-releasing intrauterine device: A mixed-method study of social franchise clients in Nigeria 

Dear Dr. Rademacher:

I'm pleased to inform you that your manuscript has been deemed suitable for publication in PLOS ONE. Congratulations! Your manuscript is now with our production department. 

Kind regards, 

on behalf of

Dr. Federico Ferrari 

Academic Editor

PLOS ONE